# Influence of Harvesting Time on the Chemical Composition of Wild Stinging Nettle (*Urtica dioica* L.)

**DOI:** 10.3390/plants10040686

**Published:** 2021-04-02

**Authors:** Aurelija Paulauskienė, Živilė Tarasevičienė, Valdas Laukagalis

**Affiliations:** Institute of Agricultural and Food Sciences, Vytautas Magnus University Agriculture Academy Studentų 11, Akademija, LT-53361 Kaunas, Lithuania; zivile.taraseviciene@vdu.lt (Ž.T.); llaukas@gmail.com (V.L.)

**Keywords:** antioxidant activity, chemical composition, harvesting time, stinging nettle

## Abstract

This research aimed to determine the effect of different harvesting times on the chemical composition of stinging nettle (*Urtica dioica* L.). The leaves of nettle were harvested at the same place once a month in the period of April–September 2019. The analysis focused on the contents of dry matter, soluble solids, ascorbic acid, titratable acidity, chlorophyll *a* and chlorophyll *b*, total carotenoids, total phenolic compounds, antioxidant activity, ash content, and macro- and microelements. The nettles harvested in April were characterized by the highest levels of soluble solids and some macro-and microelements (P, K, Fe, Zn). The plants harvested in May were distinguished for titratable acidity, chlorophyll *a*, chlorophyll *b*, and carotenoid contents. In this month, the plants were determined to have the highest antioxidant activity during the entire vegetation period. The plants collected in July contained the highest amount of Mn, but the antioxidant activity of these plants was the lowest during the vegetation period. In August, the plants had the highest levels of ascorbic acid, phenolic compounds, and ash, while the plants collected in September were characterized by having the highest amounts of Ca, Mg, and B as compared to those established in other months of vegetation.

## 1. Introduction

Stinging nettle (*Urtica dioica* L.) is a perennial herb with a long history of traditional medicinal uses in many countries. Avicenna wrote about the healing properties of nettles already a thousand years ago. *Urtica dioica* originated from the colder climate regions of northern Europe and Asia. Each year, stinging nettle produces new shoots from rhizomes and stolons, but research data show the decrease of above-ground green mass yield with crop senescence [1].

This valuable plant is characterized by the presence of numerous biologically active substances. Nettle leaves are rich in chlorophyll, carotenoids, vitamins, proteins, fats, carbohydrates, organic acids, minerals, and trace elements [2,3,4,5,6,7]. The main carotenoids established in plant leaves are β-carotene, violaxanthin, xanthophylls, zeaxanthin, luteoxanthin, and lutein epoxide [8]. This nettle has a rich vitamin composition. The plant contains both water-soluble vitamins, significant amounts of vitamin C and vitamin B (B_1_, B_2_, B_3_, B_6_, B_9_), and fat-soluble vitamins A, D, E, K, pro-vitamin A (β-carotene), and vitamin E (α-tocopherol) [5]. The leaves of stinging nettle contain abundant amounts of natural phenolic compounds, such as flavonoids, phenolic acids, anthocyanins, and other phenols, which may function as effective natural antioxidants [2,5,7,9]. Researchers report that rutin is the predominant phenolic compound in stinging nettle leaves [10,11]. Terpene diols, terpene diol glucosides, α-tocopherol, and five monoterpenoid components have also been detected in nettle leaves [12]. The *U. dioica* roots contain lectins, acidic polysaccharides, sterols, lignans, flavonoids, phytosterols, coumarins, minerals, and trace elements [5,6].

As a result of its rich chemical composition, the multidirectional therapeutic activity of this valuable plant has been known for years. Based on the literature, *U. dioica* and its phytocomponents extracted from different parts of the plant are reported to be used for various pharmacological activities, which include hypoglycaemic, anti-diabetic, anti-inflammatory, anti-rheumatic, antimicrobial, anti-asthmatic, antioxidant, diuretic, hypotensive, and analgesic activities [4,5,7,8,13,14,15,16].

In different countries, nettle is used in a variety of dishes [4,6,7,17]. In European countries, nettles are used in soup or as steamed or wilted vegetable. In Lithuania, nettle leaves are used in springtime soup. Since it has a similar flavor and texture, a cooked nettle can substitute common sorrel (*Rumex acetosa* L.) or spinach (*Spinacia oleracea* L.). Nettle leaves can also be used to make herbal tea, which is rich in vitamins, minerals, and other phytoconstituents. The results presented in one research report showed that *U. dioica* retains significant amounts of minerals, vitamins, and other functional values after heat treatment [4].

Leaves of *U. dioica* possess sharp spines with stinging hairs, which cause irritation due to formic acid, and biochemical mediators such as histamine, 5-hydroxytryptamine (serotonin), acetylcholine, and leukotrienes [7,9,16,17]. Researchers state that the mechanism of action of stinging nettles dermatitis appears to be both biochemical and mechanical [18].

The food industry uses nettles for chlorophyll extraction, and it is known as coloring agent E140 [19].

Harvesting of nettle stems up to 40 cm high with fully grown leaves starts in May–June. Stinging nettle grows well in nitrogen-rich soil, and blooms between June and September of every year [16]. Research [8] has stated that for medicinal purposes, the leaves should be picked before flowering. Picking of the plant leaves regrown after the second and third cutting can last until the first autumn frosts. The harvesting time of *U. dioica* varies depending on the method of its use. If the plant is to be used fresh, the recommended harvesting time is spring or early summer. Early-season leaves can be used as a fresh vegetable or they can be dried and used to make tea. Older leaves are not ideal for harvesting, but are generally still acceptable for making tea or tincture [8].

The aim of this research was to determine the effect of different harvesting time on the chemical composition of wild-growing stinging nettle (*Urtica dioica* L.). Our research would help to understand the chemical composition of wild nettle leaves, the amount of biologically active substances, and antioxidant activity changes during the growing season, collecting plants in the same plot all vegetation period.

## 2. Results and Discussion

### 2.1. Dry Matter and Total Soluble Solids Content in Nettle Leaves

Stinging nettle is a perennial plant with stems and leaves that emerge in spring and grow during the vegetation season. The different proportions of various compounds, as provided by the researchers, are reasoned to be caused by the plant variety, genotype, climate, soil, vegetative stage, harvest time, and other causes [5,6]. Besiada et al. [2] asserted that different amounts of dry matter (DM) are accumulated in nettles during the vegetation period. They established that leaves of the stinging nettle harvested in September contain the lowest DM content, while the highest DM quantity was obtained in May-harvest plants. Similar research results were obtained. The research data show that the highest amount of DM was in the leaves picked in June, and the lowest in September-harvested leaves (Table 1). According to scientists, such results could be influenced by the weather conditions before or at harvesting time [1]. Our results confirm this statement. During the investigation period, the month of June had a higher air temperature, exceeding the long-term average value by 3.7 °C, with an about two-fold lower rainfall as compared to the long-term average (Figure 1).

Total soluble solid (TSS) concentration in the investigated leaves of stinging nettle differed significantly (*p* < 0.05) throughout the vegetation period. The highest concentrations of TSS were found in the leaves picked in April, at the beginning of vegetation, and in September (Table 1). During the investigation period, the month of April differed in the lowest average air temperature and the lowest rainfall (Figure 1). The highest concentration of TSS in the leaves picked in April was eight-fold higher compared to that in June-harvested plants. Data analysis of our investigations showed a strong negative correlation between air temperature and nettle leaves total soluble solids (*r* = −0.78, when *p* < 0.05).

### 2.2. Ascorbic Acid Content and Titratable Acidity of Nettle Leaves

According to previous research [20], ascorbic acid in nettle leaves appears in two forms—ascorbic acid and dehydroascorbic acid. Scientists indicate that the total ascorbic acid content (AAC) may vary from 16.00 to 112.80 mg 100 g^−1^ of fresh weight (FW) or even to 238 mg 100 g^−1^ [5,20,21,22]. In our study, the AAC in nettle leaves ranged from 8.17 mg 100 g^−1^ in April to 0.58 mg 100 g^−1^ in September (Table 1). These values are lower when compared to the amounts of ascorbic acid reported by other investigators [5,20,21,22]. This could be influenced by meteorological conditions. In 2019, March (vegetation start)–September period was characterized by very low rainfall (294.90 mm), which was 1.5-fold lower than the long-term average (462.60 mm), and the average air temperature (13.34 °C) was higher than the long-term mean (12.16 °C) (Figure 1). The highest AAC was found in the leaves picked in August; this month was characterized by the highest rainfall content during the entire study period. The lowest AAC was established in the leaves harvested at the end of the vegetation period in September. In the case of investigated period ascorbic acid content in nettle leaves was not influenced by meteorological conditions.

Organic acids identified in *U. dioica* are formic, silicic, citric, fumaric, malic, oxalic, phosphoric, quinic, succinic, and threnoic acid and threono-1,4-lactone [5,23]. Organic acids are involved in the regulation of a broad range of basic cellular processes, and play a role in the control of various biochemical and physiological processes. Recent evidence even indicates that organic acids are involved with a high impact on the in vivo protein activity [24]. On the other hand, the content of short-chain organic acids in medicinal herbs is important for their taste, flavor, and therapeutic effects [25].

In our study, the titratable acidity (TA) recalculated to citric acid varied from 0.96% to 1.91% (FW) (Table 1). The highest TA was established in the plant leaves harvested in May, and the lowest in July. Data analysis of our investigations showed a strong positive correlation between growing period sunshine duration and titratable acidity of nettle leaves (*r* = 0.79, when *p* < 0.05).

### 2.3. Total Carotenoid, Chlorophyll a, and Chlorophyll b Content of Nettle Leaves

Carotenoids make another important group of the nutrients present in *U. dioica*. Nine carotenoids are identified in the leaves and found at all leaf maturity levels. Lutein, lutein isomers, β-carotene, and β-carotene isomers are the major carotenoids [26]. Some scientists have found that the total carotenoid content (TCC) varies from 5.14 to 7.48 mg 100 g^−1^ (DM), and others have stated a range of 21.6–32.3 mg 100 g^−1^ (FW) [26,27]. Our results are similar, and show a TCC range from 9.68 to 28.41 mg 100 g^−1^ (FW) (Table 1). Kriegel et al. [6] noted that higher amounts of chlorophyll and carotenoids are usually found in plants harvested in shady places. The results presented by Kukrić et al. [27] show that young nettle leaves accumulate a higher content of total chlorophyll as well as carotenoids. In our study, the amount of TCC could be influenced by the growth of nettles in a shady place and the harvesting of young leaves. We found the significantly highest TCC in May-harvested leaves, which exceeded the lowest TCC in June-harvested leaves by four-fold (Table 1). Leaves of the plants harvested in August accrued higher TCC compared to the value of other growing period months.

The main application of stinging nettle, however, is connected to obtaining chlorophyll. Nettle leaves contain a significant amount of chlorophyll, around 4.8 mg g^−1^ of dry leaves [5]. Being rich in chlorophyll, nettle leaves are used as a raw material in the pharmaceutical and food industry, as well as cosmetics production [2,27]. Researchers noted that the concentration of chlorophyll increases in growing leaves and decreases during plant ageing [2,27]. Leaves of *U. dioica* contain 69.80–158.51 mg 100 g^−1^ (FW) of chlorophyll *a* (Ch *a*) and 28.50–51.77 mg 100 g^−1^ (FW) of chlorophyll *b* (Ch *b*) depending on the climate and environmental conditions [27,28]. In the nettle leaves of our study, Ch *a* content ranged between 18.05 and 161.02 mg 100 g^−1^ (FW), while that of Ch b ranged between 6.48 and 67.69 mg 100 g^−1^ (FW) (Table 1). We determined the amounts of chlorophyll to be highly dependent on the time of harvest. The highest concentrations of chlorophyll, as well as carotenoids, were found in the plant leaves harvested in May. Ch *a* concentration was nine-fold higher and Ch *b* concentration was 10-fold higher than those in the leaves harvested in August. The nettle leaves picked in August had the lowest pigments concentration.

Carotenoids and chlorophylls are chloroplast pigments that play a vital role in plant photosynthesis processes. Carotenoids functions include light-harvesting, energy transfer, and photochemical redox reaction. Higher carotenoid content in nettle leaves may be associated with their photo-protective role [27]. Plants use green pigment chlorophyll to trap light needed for photosynthesis. The accumulation of chemical compounds by the plant tissues is a function of its genotype, temperature, sunlight, and soil fertility status [22].

The strong negative correlation between growing period temperature and pigments content (*r* = −0.85 for Ch *a*, *r* = −0.84 for Ch *b*, *r* = −0.80 for TCC, *p* < 0.05) was established. The influence of the rainfall and sunshine duration on pigments content not determined, although the sunshine duration until the middle of the growing period varied greatly.

### 2.4. Total Phenolic Content of Nettle Leaves

Studies on phytochemicals show that nettle leaves and rhizomes are rich in phenolic compounds [16,17]. The plant’s phenolic composition is affected by a variety of factors, such as genotype, climate, soil, vegetative stage of the plant, harvest time, etc. [29]. According to Grevsen et al. [29] and Besiada et al. [30], the supply of nitrogen fertilizer in the cultivation of *U. dioica* produces more plant material and may alter the concentration of phenolic compounds in aerial parts. Researchers state that higher amounts of polyphenols are accumulated in stinging nettle leaves harvested in May and July [2,30]. Our results showed the highest total phenolic content (TPC) in the nettle leaves harvested in August (Table 1). The TPC in terms of gallic acid equivalent varied between 3.28 mg GAE g^−1^ (FW) in May and 19.07 mg GAE g^−1^ (FW) in August. These results are in agreement with the results obtained by other scientists [2,29]. Polyphenol components are secondary metabolites of plants and are involved in stress protection in plants [31]. They differ in biological activity and their roles in abiotic stress responses. In our study, the highest TPC of nettle leaves accumulated in August, possibly in response to higher air temperature (Figure 1). Although the influence of meteorological conditions on the total phenolic content was not established.

### 2.5. The Antioxidant Activity of Nettle Leaves

The antioxidant activity (AA) of stinging nettle varies from 17.3 to 80.8% [7,14]. Plants harvested in different terms are characterized by markedly different antioxidant activity [30]. In our study, the AA of stinging nettle leaves differed significantly (*p* < 0.05) throughout the vegetation period, from 52.92 to 95.17% (Table 1). Significant AA was established in May-harvested leaves, which also contained the highest amounts of carotenoids and chlorophylls. Our findings confirm the researcher’s recommendations to harvest nettle leaves in spring at the beginning of the vegetation period for acquiring the highest polyphenol content and antioxidant properties [10,30,31]. Some authors suggest that the antioxidant activity of nettle leaves is strongly correlated with the phenolic compounds and their structure [5,7,9,17,30,31,32]. According to other authors, the antioxidant activity of nettle is determined not only by phenols but also by ascorbic acid, chlorophylls, carotenoids, and other compounds [22,24,25,33]. A data analysis of our investigations showed a moderate positive correlation between nettle leaves antioxidant activity and Ch *a* (*r* = 0.67), Ch *b* (*r* = 0.56), TCC (*r* = 0.56), and TPC (*r* = 0.62) when *p* < 0.05.

### 2.6. Crude Ash and Mineral Elements in Nettle Leaves

The content of mineral substances in nettle is about 20% of the dry mass [5]. The researchers found that the amount of crude ash (CA) in nettle leaves can range from 2.1 to 3.4% [4,22]. In our investigation, the CA content detected in nettle leaves was similar to the findings of some authors and ranged between 3.06% at the beginning of the vegetation period to 3.73% at the end of vegetation (Table 2). CA content increased steadily until August when its peak value was set and decreased in September.

Stinging nettle is a valuable source of macro and microelements [2,5,21,22]. Literature sources give different proportions of various compounds. According to researchers, the amount of potassium in nettle leaves can vary between 1.27% [22] to 2.27% [21], and the amount of calcium from 2.14% [22] to 2.63% [3] or even to 5.21% [21] of DM. Shonte et al. [22] found that nettle leaves contain about 0.55% of phosphorus and 0.69% of magnesium, while Rafajlovska et al. [3] specified 2.51% of magnesium. Researchers have shown that nettles are rich in iron, but report different amounts, ranging from 167 mg kg^−1^ [22] to 2765 mg kg^−1^ [21] in the plant leaves. Nettle leaves may contain about 12.26 mg kg^−1^ of copper [3]; moreover, leaves may have zinc contents from 27.44 mg kg^−1^ [3] to 35.00 mg kg^−1^ [22] and manganese contents from 4.03 mg kg^−1^ [3] to 25.00 mg kg^−1^ [22] of DM. In nettle, the concentration of mineral elements varies strongly because of the different absorption of mineral elements from the soil and the time of sample collection [21,22]. Research results show that high contents in calcium, magnesium, phosphorus, iron, and manganese allow classifying nettle plant among the food able to bring supplements in these elements [4,5,22]. A typical serving of fresh or dried nettle leaves provides approximately 5 to 30% of the recommended daily values of some mineral elements [22].

The concentrations of nine mineral elements (P, K, Ca, Mg, Fe, Cu, Mn, Zn, B) determined in nettle leaves is shown in Table 2. Our results are in agreement with the findings of other authors [2,3,5,6,21,22,33]. In our study, the highest amounts of potassium (average 3.18%) and calcium (average 3.04%) were found in the nettle leaves. Phosphorus content (average 0.82%) was about four-fold lower and magnesium content (average 0.61%) was six-fold lower. Regarding the microelements, the highest values were reached by iron (average 224.78 mg kg^−1^), followed by boron (average 49.25 mg kg^−1^) and manganese (average 44.65 mg kg^−1^). The lowest concentration was identified for zinc (average 18.86 mg kg^−1^) and copper (average 14.23 mg kg^−1^). Radman et al. [21] stated that high contents of iron and potassium in nettle can be the result of high levels of these minerals in the soil. Our results confirm these statements. According to the results of the chemical analysis of our experiment, the soil was rich in available iron (1324 mg kg^−1^) and potassium (655 mg kg^−1^).

The highest concentrations of phosphorus, potassium, iron, and zinc were in the nettle leaves harvested in April, while the highest concentration of calcium, magnesium, and boron were in the nettles harvested in September, at the end of vegetation (Table 2). Copper and manganese concentrations increased from springtime to the middle of the vegetation period and reached the highest concentrations of copper in June and of manganese in July. Concentrations of other mineral elements fluctuated during the vegetation period.

## 3. Materials and Methods

### 3.1. Plant Material Collection and Soil Properties

The nettle plants were harvested at the same place in the central region of Lithuania (54°53′ N, 23°50′ E) once a month in the period of April–September 2019. The plant collection was done in a plot, located away from cultivated agricultural fields, industrial enterprises, and highways. The area of the experimental plot was 3 m^2^. The plot was divided into three fields of 1 m^2^ (three replications). One kg of nettle leaves was harvested separately from each field and laboratory samples were prepared.

The soil of experimental field was limnoglacial loam, deposited moraine loam, and deeper carbonate gleyic luvisol (IDg p1/p2). The main properties of the soil plough layer were as follows: pH_KCl_ 6.4, available phosphorus (P_2_O_5_)—362 mg kg^−1^, available potassium (K_2_O)—655 mg kg^−1^, total nitrogen (N)—0.26%, available iron (Fe)—1324 mg kg^−1^.

### 3.2. Meteorological Conditions

In 2019, the air temperature was higher than the long-term mean value (Figure 1). In March, when nettle vegetation began, the long-term average air temperature was exceeded by more than 3 °C, while in April more than 2 °C. Higher temperature values were observed in June, August, and October. Moreover, rainfall was lower than the long-term average. Though in March the rainfall was higher by 6 mm, during all other months of vegetation the rainfall was lower, by 40.7 to 20.7 mm.

The sunshine duration varied in the decades of each month. At the beginning of the nettle growing period in the first and second decades of March, the sun shone for a few hours (Figure 2). Only in the third decade of March, the sunshine duration lengthened. April and June were distinguished by the number of hours of sunshine for all decades. In May, July, and August, the sun was shining an average number of hours. The sunshine of the first decade of September was equal to the third decade of August. Moreover, from the second decade of September, the sunshine duration started to shorten.

### 3.3. Chemical Analysis

Chemicals used in this study were of analytical grade. Chemical analyses were performed in three replications.

Determination of chemical composition was based on the use of fresh leaves of stinging nettle plants. The chemical composition of leaves was examined once a month in the period of April to September. The leaves were analyzed immediately after harvesting. For chemical analysis, a 1-kg laboratory sample was made of homogenized leaves. Chemical analyses were performed in three replications.

#### 3.3.1. Chemical Analysis of Soil

The soil pH_KCl_ was measured in 1N KCl extraction by potentiometric method [34]; available phosphorus (P_2_O_5_) and available potassium (K_2_O) were measured by the Egner–Riehm–Domingo (A–L) method [35]. Total nitrogen content in the soil samples was determined by the Kjeldalh method [36]. Available iron (Fe) was established by inductively coupled plasma atomic emission spectrometry (ICP-AES).

#### 3.3.2. Dry Matter and Total Soluble Solids Content

Dry matter (DM) content was assessed by drying the leaves samples to a constant mass at 105 °C [37]; soluble solids concentration (TSS) was measured in the juice pressed from homogenized nettle leaves by using a digital refractometer PAL-1 (Atago, Japan) at 20 °C [37].

#### 3.3.3. Ascorbic Acid Content and Titratable Acidity

Ascorbic acid content (AAC) was determined by titration with 2,6-dichlorphenol-indophenol sodium salt dehydrate [37]. Titratable acidity (TA) was determined by titration with 0.1 N sodium hydroxide solution to the definite pink endpoint and expressed as a percentage of citric acid [37].

#### 3.3.4. Total Carotenoid, Chlorophyll a, and Chlorophyll b Content

Total carotenoid content (TCC) and the chlorophyll *a* (Ch *a*) and chlorophyll *b* (Ch *b*) contents were analyzed by using a two-ray UVS-2800 spectrophotometer (Labomed Inc., Los Angeles, CA, USA). The absorbance was read at 662 nm (for chlorophyll *a*), 645 nm (for chlorophyll *b*), and 470 nm (for total carotenoids content) and the number of pigments was calculated as described by Straumite et al. [38]. For pigment extraction, a 0.4-g sample of homogenized leaves was weighed in a conical flask, extracted with 10 mL of 100% acetone, and mixed using a magnetic stirrer (VWR, USA) for 15 min at 700 rpm. The supernatant was then separated and the extraction was repeated. The extraction process was done in triplicate [38].

#### 3.3.5. Total Phenolic Content

The total phenolic content (TPC) was established in the leaves by using the Folin–Ciocalteu reagent. Ten ml of ethanol (70 %) was added to a 0.25 g sample of homogenized leaves and extracted in the ultrasonic bath in the 20 °C water for 30 min. Then, the extract was centrifuged at 3000 rpm for 30 min. Then, 0.2 mL of prepared extract was mixed with 1 mL of the Folin–Ciocalteu reagent, after which 0.8 mL of sodium carbonate (7%) was added and the mixture was completed to 5 mL with pure water. After 60 min of incubation at 20 °C in the dark, the absorbance was measured at 760 nm using a two-ray UVS-2800 spectrophotometer (Labomed Inc., Los Angeles, CA, USA). Total phenolic content was measured with the calibration curve by using gallic acid equivalent standards. The results were expressed as equivalents of gallic acid (mg GAE g^−1^) [9].

#### 3.3.6. Antioxidant Activity

The antioxidant activity (AA) was determined by using the 2.2-diphenyl-1-picrylhydrazyl free radical (DPPH) method. One g of homogenized nettle leaves was mixed with 10 mL of 95% methanol and shaken for 30 min, then centrifuged for 15 min at 3000 rpm. The amount of 3 mL 0.1 mM of DPPH methanol solution was added to the 0.3 mL of an aliquot of the methanolic extract. The absorbance was read at the 515 nm wavelength by using a two-ray UVS-2800 spectrophotometer (Labomed Inc., Los Angeles, CA, USA) [14].

#### 3.3.7. Crude Ash and Mineral Elements

The amount of crude ash (CA) was determined by the dry burning of samples at a temperature of 500 °C [39]. The number of mineral elements was determined from the dry mass (DM) of the nettle leaves. Nitrogen (N) was identified by the Kjeldahl method. Phosphorus (P) was measured photometrically [40]. Calcium (Ca) content was established according to Commission Directive 71/250/EEC [39]. The ash was treated with hydrochloric acid and the calcium precipitated as calcium oxalate. The precipitate was dissolved in sulfuric acid and the formed oxalic acid was titrated with potassium permanganate solution. The ash was dissolved in hydrochloric acid, after which the potassium (K) content of the solution was determined by flame photometry [39]. Magnesium (Mg), iron (Fe), copper (Cu), manganese (Mn), zinc (Zn), and boron (B) contents were identified by atomic absorption spectrophotometry [41].

### 3.4. Statistical Analysis

Data analysis was carried out with STATISTICA version 7 software (TIBCO Software, Palo Alto, CA, USA). The results were analyzed by using a one-way analysis of variance (ANOVA). Arithmetical means and standard deviations of the experimental data were calculated. Fisher’s Least-Significant-Difference test (LSD) was applied to the experimental results to assess significant differences between mean values at a significance level of *p* < 0.05. Linear correlation analysis was performed to determine the strength and character of the relationships between variables at a probability level of 95%.

## 4. Conclusions

Our research showed that wild-growing nettle (*Urtica dioica* L.) leaves can be harvested from April to September in the same plot due to accumulated chemical compounds. The chemical composition of nettle leaves varied throughout the growing season. The highest concentrations of TSS, phosphorus, potassium, iron, and zinc were found in the leaves picked in April, at the beginning of vegetation. The leaves harvested in May had the highest titratable acidity, chlorophyll *a*, chlorophyll *b*, and carotenoid content. The highest antioxidant activity of plants was in May. Data analysis showed a moderate positive correlation between antioxidant activity and chlorophyll *a*, chlorophyll *b*, carotenoids, and phenols this period. In June, the highest dry matter and copper contents were found in leaves. The leaves harvested in July were characterized by the highest crude ash and manganese contents, but the antioxidant activity was the lowest one for the entire vegetation period. In August, the highest levels of ascorbic acid, phenolic compounds, and crude ash were determined in nettle leaves. Crude ash content did not differ significantly in July and August. In September, nettle leaves contained the most calcium, magnesium, and boron as compared to the values established in other months of vegetation.

The obtained research data showed that nettle leaves chemical composition probably depends on meteorological conditions, and for this reason, may vary each year in the same months.

## Figures and Tables

**Figure 1 plants-10-00686-f001:**
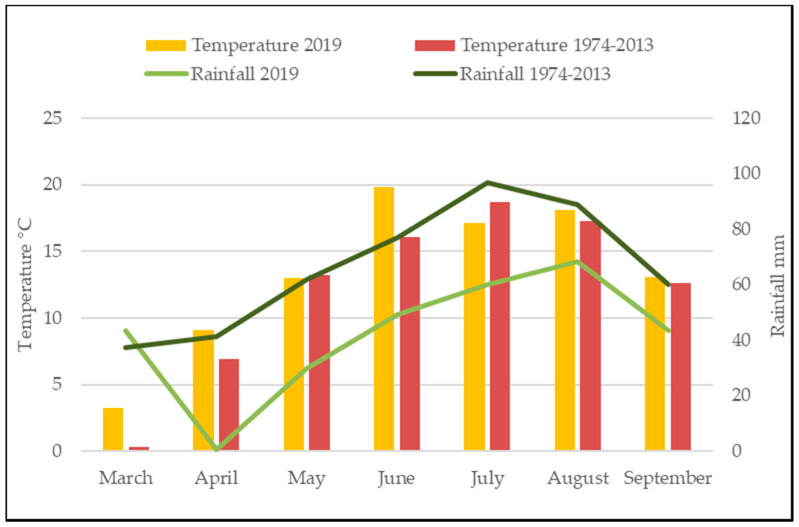
Average temperature and rainfall sum during nettle growing period (Kaunas meteorological station).

**Figure 2 plants-10-00686-f002:**
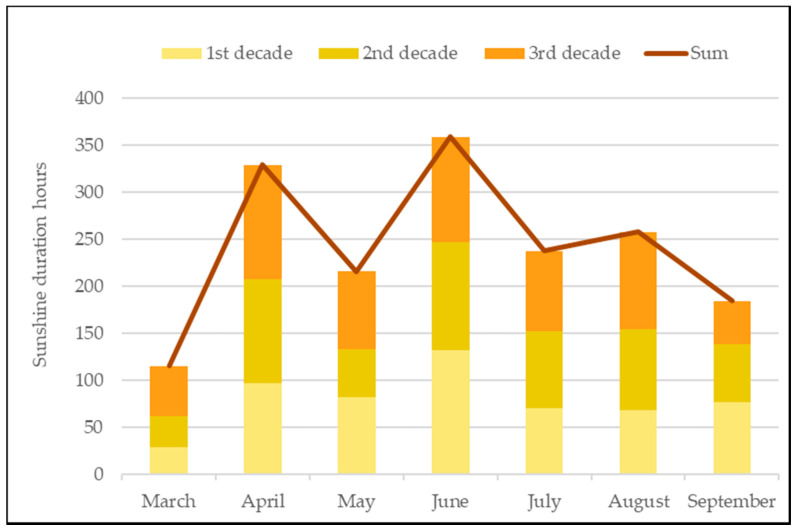
Sunshine duration in nettle growing period (Kaunas meteorological station).

**Table 1 plants-10-00686-t001:** Chemical composition of stinging nettle (*Urtica dioica* L.) leaves (FW).

	April	May	June	July	August	September
Dry matter, %	21.24 ± 0.53 ^bc^*	21.57 ± 0.15 ^bc^	24.41 ± 0.15 ^a^	21.13 ± 0.18 ^bc^	22.61 ± 0.53 ^ab^	20.48 ± 0.43 ^c^
Total soluble solids, %	11.70 ± 0.49 ^a^	3.47 ± 0.26 ^d^	1.47 ± 0.09 ^f^	4.60 ± 0.15 ^c^	2.57 ± 0.03 ^e^	8.07 ± 0.26 ^b^
Ascorbic acid. mg 100 g^−1^	8.17 ± 0.59 ^b^	6.45 ± 0.29 ^c^	1.31 ± 0.25 ^d^	9.02 ± 0.29 ^b^	25.66 ± 1.04 ^a^	0.58 ± 0.15 ^d^
Total carotenoids, mg 100 g^−1^	12.68 ± 0.15 ^cd^	28.41 ± 0.45 ^a^	6.46 ± 0.82 ^e^	15.70 ± 0.79 ^c^	19.52 ± 0.89 ^b^	9.68 ± 0.86 ^d^
Chlorophyll *a*, mg 100 g^−1^	40.47 ± 0.59 ^d^	161.02 ± 1.86 ^a^	18.05 ± 0.41 ^e^	64.31 ± 1.01 ^c^	80.46 ± 2.15 ^b^	39.43 ± 1.71 ^d^
Chlorophyll *b*, mg 100 g^−1^	16.40 ± 0.95 ^c^	67.69 ± 1.96 ^a^	6.48 ± 0.15 ^d^	32.60 ± 0.74 ^b^	36.97 ± 1.74 ^b^	14.67 ± 1.46 ^cd^
Titratable acidity, %	1.27 ± 0.32 ^ab^	1.91 ± 0.28 ^a^	1.10 ± 0.01 ^b^	0.96 ± 0.16 ^b^	1.60 ± 0.16 ^ab^	1.43 ± 0.27 ^ab^
Total phenolics, mg GAE g^−1^	7.96 ± 0.02 ^c^	3.28 ± 0.76 ^e^	3.62 ± 0.01 ^e^	5.32 ± 0.01 ^d^	19.07 ± 0.92 ^a^	13.96 ± 0.02 ^b^
Antioxidant activity, %	63.35 ± 0.06 ^d^	95.17 ± 0.01 ^a^	57.32 ± 0.01 ^e^	52.92 ± 0.30 ^f^	68.99 ± 0.23 ^c^	84.44 ± 0.01 ^b^

* Significant differences (*p* < 0.05) in lines are marked by different letters; for each measured parameter, the general mean ± SD is presented; FW—fresh weight; mg GAE g^−1^—results expressed as equivalents of gallic acid.

**Table 2 plants-10-00686-t002:** Crude ash and mineral elements of stinging nettle (*Urtica dioica* L.) leaves (DM).

	April	May	June	July	August	September
Crude ash, %	3.06 ± 0.10 ^c^*	3.14 ± 0.07 ^c^	3.48 ± 0.02 ^bc^	4.37 ± 0.14 ^a^	4.70 ± 0.20 ^a^	3.73 ± 0.19 ^b^
Phosphorus, %	1.02 ± 0.01 ^a^	0.83 ± 0.01 ^b^	0.76 ± 0.01 ^d^	0.72 ± 0.01 ^e^	0.80 ± 0.01 ^bc^	0.77 ± 0.02 ^cd^
Potassium, %	3.60 ± 0.01 ^a^	3.51 ± 0.01 ^b^	2.98 ± 0.01 ^d^	2.59 ± 0.01 ^e^	3.41 ± 0.01 ^c^	2.98 ± 0.00 ^d^
Calcium, %	2.21 ± 0.01 ^f^	2.85 ± 0.01 ^d^	2.82 ± 0.01 ^e^	3.32 ± 0.02 ^b^	3.05 ± 0.01 ^c^	3.97 ± 0.01 ^a^
Magnesium, %	0.42 ± 0.01 ^f^	0.46 ± 0.01 ^d^	0.67 ± 0.01 ^b^	0.68 ± 0.02 ^b^	0.60 ± 0.01 ^c^	0.81 ± 0.01 ^a^
Iron, mg kg^−1^	526.20 ± 0.12 ^a^	148.37 ± 0.09 ^e^	172.53 ± 0.09 ^c^	165.40 ± 0.15 ^d^	112.60 ± 0.12 ^f^	223.60 ± 0.12 ^b^
Copper, mg kg^−1^	13.40 ± 0.15 ^d^	17.53 ± 0.09 ^b^	18.43 ± 0.12 ^a^	13.87 ± 0.09 ^c^	11.90 ± 0.12 ^e^	10.23 ± 0.09 ^f^
Manganese, mg kg^−1^	40.90 ± 0.06 ^e^	48.13 ± 0.09 ^c^	48.87 ± 0.15 ^b^	57.40 ± 0.06 ^a^	30.50 ± 0.06 ^f^	42.11 ± 0.02 ^d^
Zinc, mg kg^−1^	34.30 ± 0.12 ^a^	17.93 ± 0.09 ^b^	16.60 ± 0.12 ^d^	17.40± 0.06 ^c^	12.70 ± 0.12 ^f^	14.20 ± 0.12 ^e^
Boron, mg kg^−1^	40.07 ± 0.12 ^e^	31.93 ± 0.15 ^f^	50.20 ± 0.12 ^d^	58.40 ± 0.12 ^b^	52.77 ± 0.09 ^c^	62.13 ± 0.09 ^a^

* Significant differences (*p* < 0.05) in lines are marked by different letters; for each measured parameter, the general mean ± SD is presented.

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
