# Peer review of "Influence of Harvesting Time on the Chemical Composition of Wild Stinging Nettle (Urtica dioica L.)"

_plants, 2021, doi:10.3390/plants10040686_

Round 1
Reviewer 1 Report
Dear Authors,
I am sending a review of the article “Influence of Harvesting Time on the Chemical Composition of Wild Stinging Nettle (Urtica dioica L.)”.
Title “Influence of Harvesting Time on the Chemical Composition of Wild Stinging Nettle (Urtica dioica L.)” is adequate to the content. The aim of this research was to determine the effect of different harvesting time on the chemical composition of stinging nettle (Urtica dioica L.). Purpose of research has been achieved, but the manuscript needs to be improved.
Introduction
The Authors did not provide arguments for the need for research. Such research topics have already appeared in the literature. The Authors did not provide information on what new their research would bring to science.
This section should be improved.
Line 29 … U. dioica originally …
Sentence beginning – maybe a better start Stinging nettle?
Line 56 … common sorrel (Rumex acetosa) or spinach (Spinacia oleracea) …
Full latin name should be given.
Line 87 … DM …
Acronym for the first time – full name should be given.
Line 97 … TSS …
Acronym for the first time – full name should be given.
Line 107 … According to researchers [25], …
The literature should be numbered from this place,– lack 20-24 – citied in secion Materials and Methods.
Line 108 … AAC …
Acronym for the first time – full name should be given.
Line 109 … FW …
Acronym for the first time – full name should be given.
Line 114 … rainfall (294.90 mm), which was 1.6-fold lower than the long-term average (462.60 mm) …
Rainfall data – too accurately.
Line 115
Temperature unit should be corrected.
Line 126 … TA …
Acronym for the first time – full name should be given.
Line 137 … TCC …
Acronym for the first time – full name should be given.
All acronyms for the first time – full name should be given.
Line 193-195 … Data analysis of our investigations shows a moderate positive correlation 193 between nettle leaves antioxidant activity and Ch a (r = 0.67, R2 = 0.45), Ch b (r = 0.56, R2= 194 0.33), TCC (r = 0.56, R2 = 0.32) and TPC (r = 0.62, R2 = 0.38) when p < 0.05. …
R2 values are unnecessary, they do not bring new information.
Regression analysis should be removed.
Materials and Methods
Line 237-239 … The nettle plants were harvested at the same place in the central region of Lithuania (54°53′ N, 23°50′ E) once a month in the period of April–September 2019. The plant collection was done in the plot, located away from cultivated agricultural fields, industrial enterprises and highways. …
This part should be completed: 1) number of plots, 2) area, 3) number of plants, 4) number of samples, 5) weight of samples, etc.
Line 241-244 … The soil was limnoglacial loam deposited moraine loam, deeper carbonate gleyic luvisol (IDg p1/p2). The main properties of the soil plough layer were as follows: pHKCl 6.4, 242 available phosphorus (P2O5) – 362 mg kg-1, available potassium (K2O) – 655 mg kg-1, total 243 nitrogen (N) – 0.26%, available iron (Fe) – 1324 mg kg-1.
These data should be included in the Results and Discussion section – there are own data.
Line 256 … Figure 1. Average temperature and rainfall sum during nettle vegetation period (Kaunas meteorological station) …
There are presented data not only for nettle vegetation period – needs to be completed.
Line 265-267 … The leaves were analysed immediately after harvesting. For chemical analysis, a 1 kg laboratory sample was made of homogenised leaves. Chemical analyses were performed in three replications. …
Methodical error – when leaves were good homogenised, SD value will show measurement error, but not variation in plant material. Three separate leaves samples should be taken and separatelly analysed.
Line 343 … level of p < 0.05. Linear correlation and regression analyses were performed to determine …
Regression analyses is unnecessary, they do not bring new information.
Conclusions
This section presents the results of the research. At this point, the conclusions are important, namely: what follows from these results; needs improvement.
References
Line 369 … Changes in the productivity of wild and cultivated … ; Line 376 … Mineral Properties and Dietary Value of Raw and …
Which form of the record is correct – please standardize it throughout the text.
Line 372 … Stinging Nettle (Urtica Dioica L.).
Requires correction – standardize the spelling in the text.
Line 374 … Rafajlovska, V., Kavrakovski, Z., Siminovska, J., Srbinoska, M. …
Requires correction.
Line 374 …stinging nettle …
Requires correction – standardize the spelling in the text.
Line 374 … Rutto, L. K.;
Space between L. and K. or no? – standardize in the text.
Line 377 … 2013, 2013, …
???
Author Response
Thank you for the evaluation of our article.
Article corrections have been made.
1. Introduction
The Authors did not provide arguments for the need for research. Such research topics have already appeared in the literature. The Authors did not provide information on what new their research would bring to science.
This section should be improved.
A broader explanation of the purpose of the study was inserted, lines 78-81.
2. Line 29 … U. dioica originally …
Sentence beginning – maybe a better start Stinging nettle?
To avoid a repetition of "stinging nettle" in all sentences of the introduction first paragraph, in the second sentence was written a Latin name Urtica dioica.
3. Line 56 … common sorrel (Rumex acetosa) or spinach (Spinacia oleracea) …
Full latin name should be given.
Corrections were made.
4. Line 87 … DM …
Acronym for the first time – full name should be given.
Line 97 … TSS …
Acronym for the first time – full name should be given.
Corrections were made.
5. Line 107 … According to researchers [25], …
The literature should be numbered from this place,– lack 20-24 – citied in secion Materials and Methods.
Literature numbering were corrected.
6. Line 108 … AAC …
Acronym for the first time – full name should be given.
Line 109 … FW …
Acronym for the first time – full name should be given.
Corrections were made.
7. Line 114 … rainfall (294.90 mm), which was 1.6-fold lower than the long-term average (462.60 mm) …
Rainfall data – too accurately.
Was corrected to 1.5-fold.
8. Line 115
Temperature unit should be corrected.
Was corrected.
9. Line 126 … TA …
Acronym for the first time – full name should be given.
Line 137 … TCC …
Acronym for the first time – full name should be given.
All acronyms for the first time – full name should be given.
Corrections were made.
10. Line 193-195 … Data analysis of our investigations shows a moderate positive correlation 193 between nettle leaves antioxidant activity and Ch a (r = 0.67, R2 = 0.45), Ch b (r = 0.56, R2= 194 0.33), TCC (r = 0.56, R2 = 0.32) and TPC (r = 0.62, R2 = 0.38) when p < 0.05. …
R2 values are unnecessary, they do not bring new information.
Regression analysis should be removed.
Corrections were made, regression coefficients were removed.
11. Materials and Methods
Line 237-239 … The nettle plants were harvested at the same place in the central region of Lithuania (54°53′ N, 23°50′ E) once a month in the period of April–September 2019. The plant collection was done in the plot, located away from cultivated agricultural fields, industrial enterprises and highways. …
This part should be completed: 1) number of plots, 2) area, 3) number of plants, 4) number of samples, 5) weight of samples, etc.
The information in the “Materials and Methods“ section was added, lines 247-249.
12. Line 241-244 … The soil was limnoglacial loam deposited moraine loam, deeper carbonate gleyic luvisol (IDg p1/p2). The main properties of the soil plough layer were as follows: pHKCl 6.4, 242 available phosphorus (P2O5) – 362 mg kg-1, available potassium (K2O) – 655 mg kg-1, total 243 nitrogen (N) – 0.26%, available iron (Fe) – 1324 mg kg-1.
These data should be included in the Results and Discussion section – there are own data.
Soil characteristic is usually given in the “Materials and Methods” section, for this reason, we left this data in the mentioned section.
Examples:
Jankauskienė, Z.; Gruzdevienė, E. Changes in the productivity of wild and cultivated stinging nettle (Urtica dioica L.) as influenced by the planting density and crop age. Zemdirbyste 2015, 102, 31–40. https://doi.org/10.13080/z-a.2015.102.004.
Biesiada, A.; Kurcharska, A.; Sokól-Lêtowska, A.; Kus, A. Effect of the age of plantation and harvest term on chemical composition and antioxidant avctivity of Stinging Nettle (Urtica Dioica L.). Ecol. Chem. Eng. 2010, 17, 1061–1067. Available online: bwmeta1.element.baztech-article-BPG8-0060-0014.
Zeipina, S.; Alsina, I.; Lepse, L.; Dūma, M. Antioxidant activity in nettle (Urtica dioica L.) and garden orache (Atriplex hortensis L.) leaves during vegetation period. Cheminė technologija 2015, 1, 29–33. https://doi.org/10.5755/j01.ct.66.1.12055
13. Line 256 … Figure 1. Average temperature and rainfall sum during nettle vegetation period (Kaunas meteorological station) …
There are presented data not only for nettle vegetation period – needs to be completed.
The title of Figure 1 changed from "...vegetation period" to "...growing period".
The nettles started to grow in March but the plants were so small that we couldn't collect enough for the analyses. But we think that the temperatures, rainfall and sunshine duration in March affected the chemical composition of the leaves harvested in April.
14. Line 265-267 … The leaves were analysed immediately after harvesting. For chemical analysis, a 1 kg laboratory sample was made of homogenised leaves. Chemical analyses were performed in three replications. …
Methodical error – when leaves were good homogenised, SD value will show measurement error, but not variation in plant material. Three separate leaves samples should be taken and separatelly analysed.
The information in the “Materials and Methods“ section about plant material collection was added, lines 247-249.
15. Line 343 … level of p < 0.05. Linear correlation and regression analyses were performed to determine …
Regression analyses is unnecessary, they do not bring new information.
Was corrected.
16. Conclusions
This section presents the results of the research. At this point, the conclusions are important, namely: what follows from these results; needs improvement.
Conclusions were supplemented, line 365-366 and line 380-382.
17. References
Line 369 … Changes in the productivity of wild and cultivated … ; Line 376 … Mineral Properties and Dietary Value of Raw and …
Which form of the record is correct – please standardize it throughout the text.
Line 372 … Stinging Nettle (Urtica Dioica L.).
Requires correction – standardize the spelling in the text.
Line 374 … Rafajlovska, V., Kavrakovski, Z., Siminovska, J., Srbinoska, M. …
Requires correction.
Line 374 …stinging nettle …
Requires correction – standardize the spelling in the text.
Line 374 … Rutto, L. K.;
Space between L. and K. or no? – standardize in the text.
Line 377 … 2013, 2013, …
???
All inaccuracies in the “References” section have been corrected.
Reviewer 2 Report
This manuscript is a descriptive work indicating the effect of time of harvesting and chemical composition of nettle leaves. The results lack of originality and has a local interest. No important correlations are found between the reported data and the parameters considered, which suppose a general interest for readers worlwide. In addition, open field experiments studing plants developments shouls consider light intensity, or the hours of ligkt a day, particularly if you are measuring pigment contents in leaves.
Author Response
This manuscript is a descriptive work indicating the effect of time of harvesting and chemical composition of nettle leaves. The results lack of originality and has a local interest. No important correlations are found between the reported data and the parameters considered, which suppose a general interest for readers worlwide. In addition, open field experiments studing plants developments shouls consider light intensity, or the hours of ligkt a day, particularly if you are measuring pigment contents in leaves.
We grateful reviewer for the comments.
We hope our research data will be interesting for readers in other countries as well.
There are a number of studies about nettle growing and fertilization, the influence of fertilizers on the plants chemical composition, changes in chemical composition during leaves, stalks and roots drying, etc., about the pharmacological and therapeutic properties of nettles. However, there is a lack of information about changes in the chemical composition of wild-growing plants when the nettles are harvested in the same plot all growing season.
We have not measured light intensity and cannot provide such data. But the “Materials and Methods” section was supplemented with Figure 2. Sunshine duration in nettle growing period.
Correlation analysis between pigments in investigated nettle leaves and meteorological conditions was performed but correlation was found only between pigments and vegetation period air temperature. Line 167-176.
Authors.
Reviewer 3 Report
The present manuscript aims at studying the influence of harvesting month on the chemical composition and antioxidant activity of stinging nettle (Urtica dioica L.) leaves. The authors have observed large variations in the organic compounds and minerals contents, as well as slightly high alterations in the DPPH scavenging effect of stinging nettle leaves extracts, during May-September period, which may be affected by climatic changes and soil properties. Nevertheless, there are some aspects that the authors should take in consideration for improving the present manuscript, in order to meet the Journal’s requirements. Please, take in account the following specific comments and questions.
Specific comments:
1 – Lines 34-40; 52-63: the references 5-8, 13, 16 and 17 represent review papers. The authors should consider in including more research papers instead of review ones.
2 – Lines 77-78: the authors should add the antioxidant activity in the aim of the present manuscript.
3 – Lines 80-195: the authors should explain the abbreviations at the first time that they were referred in the body text, e.g. DM (line 87), TSS (line 97), AAC (line 108), TA (line 106), FW (line 138), TCC (line 142) and TPC (line 172).
4 – Lines 97-103; 112-118; 126-128: the authors need to discuss more deeply some results, and consequently to support more the discussion by comparing them with literature data, for instance, the influence of climatic conditions on the TSS, AAC and TA.
5 – Lines 199-203; 212-214: Regarding crude ash and mineral elements data, the authors did not expose the respective concentration ranges described in the literature.
6 – References’ order: the authors should organize the references’ order along the manuscript. For instance, the reference 19 was in line 67, whilst the reference 20 appeared in line 285, and then the references 24 and 25 were in lines 334 and 107, respectively.
7 – Lines 237-240: the plants’ age should be mentioned.
8 – Figure 1: Why was the 1974-2013 time range chosen?
9 – Lines 269-320: Several references are lacking in the experimental procedure concerning the soil chemical analysis (lines 269-275), total soluble solids content (lines 279-280), titratable activity (lines 285-287), pigments extraction (lines 295-298) and DPPH scavenging effect (lines 313-320).
10 – Lines 300-311: Why did the authors remove phenolic compounds from stinging nettle leaves, by ultrasound-assisted extraction and using 70% ethanol as the extraction solvent? Were the TPC expressed as mg GAE g-1 fresh weight or mg GAE g-1 extract?
11 – Lines 313-320: The concentration of DPPH methanol solution should be described, and the symbol (*) should be removed. Why did not the authors determine the DPPH scavenging effect in the stinging nettle leaves extracts prepared in the section 3.3.5? DPPH scavenging effect of extracts prepared in the section 3.3.6 may be different relatively to the formers. For the antioxidant activity, more tests should be approached, e.g. reducing power and/or ABTS scavenging effect.
Author Response
Specific comments:
1 – Lines 34-40; 52-63: the references 5-8, 13, 16 and 17 represent review papers. The authors should consider in including more research papers instead of review ones.
Review papers summarize represent the research of many scientists. For this reason, we believe that these literature sources are fit for the "Introduction" section where usually provides common information about the research object and the aim of the research. We based it in part on information from review papers, as well as information from research papers, the references 2,3,4,9,10,11,14,15,18.
2 – Lines 77-78: the authors should add the antioxidant activity in the aim of the present manuscript.
The explanation of the aim of the research has been expanded, line 78-81.
3 – Lines 80-195: the authors should explain the abbreviations at the first time that they were referred in the body text, e.g. DM (line 87), TSS (line 97), AAC (line 108), TA (line 106), FW (line 138), TCC (line 142) and TPC (line 172).
Corrections were made.
4 – Lines 97-103; 112-118; 126-128: the authors need to discuss more deeply some results, and consequently to support more the discussion by comparing them with literature data, for instance, the influence of climatic conditions on the TSS, AAC and TA.
There are a number of studies about nettle growing and fertilization, the influence of fertilizers on the plants chemical composition, changes in chemical composition during leaves, stalks and roots drying, etc, about the pharmacological and therapeutic properties of nettles. However, there is a lack of usually available scientific information about changes in the chemical composition of wild-growing plants depending on climate conditions. On the other hand, TSS and TA are generalized parts of the chemical composition that are rarely presented in research articles.
Correlation analysis between meteorological conditions and nettle leaves TSS, AAC and TA was performed but correlation was found only between air temperature and nettle leaves TSS and between sunshine duration and nettle leaves TA, lines 107-108 and 134-136.
5 – Lines 199-203; 212-214: Regarding crude ash and mineral elements data, the authors did not expose the respective concentration ranges described in the literature.
In lines 226-234 were indicated mineral concentrations described in the literature.
6 – References’ order: the authors should organize the references’ order along the manuscript. For instance, the reference 19 was in line 67, whilst the reference 20 appeared in line 285, and then the references 24 and 25 were in lines 334 and 107, respectively.
References discrepancies were corrected.
7 – Lines 237-240: the plants’ age should be mentioned.
Since wild-growing nettle plants were selected for the study, we cannot indicate the plants’ age. For complete clarity, the purpose of the research has been supplemented, line 78.
8 – Figure 1: Why was the 1974-2013 time range chosen?
This time range is not our choice. Such long-term data are presented by the meteorological station.
9 – Lines 269-320: Several references are lacking in the experimental procedure concerning the soil chemical analysis (lines 269-275), total soluble solids content (lines 279-280), titratable activity (lines 285-287), pigments extraction (lines 295-298) and DPPH scavenging effect (lines 313-320).
References were added.
10 – Lines 300-311: Why did the authors remove phenolic compounds from stinging nettle leaves, by ultrasound-assisted extraction and using 70% ethanol as the extraction solvent? Were the TPC expressed as mg GAE g-1 fresh weight or mg GAE g-1 extract?
Analysis of phenolic compounds was performed according methodology described by the Otles and Yalcin [9]. Ultrasound-assisted extraction using 70% ethanol performed the best extraction of phenolic compounds.
Chemical composition analysis was performed on fresh nettle leaves and it is described in the “Materials and Methods” section, line 295-299. The TPC expressed as mg GAE g-1 of fresh weight (Table 1).
11 – Lines 313-320: The concentration of DPPH methanol solution should be described, and the symbol (*) should be removed. Why did not the authors determine the DPPH scavenging effect in the stinging nettle leaves extracts prepared in the section 3.3.5? DPPH scavenging effect of extracts prepared in the section 3.3.6 may be different relatively to the formers. For the antioxidant activity, more tests should be approached, e.g. reducing power and/or ABTS scavenging effect.
DPPH analysis was performed according methodology described by the Zeipina et al. [14]. The symbol (*) was removed and the concentration of methanol was specified.
Thank you for your comments. However, we will only be able to perform more tests for antioxidant activity in future research.
We grateful reviewer for the comments.
Authors
Reviewer 4 Report
The presented manuscript is very well planned and written. The biochemical and physiological side of these studies does not raise any substantive or methodological concerns. The authors clearly presented their argument throughout the text. The structure of the text is good.
Attention should be paid to the very high utilitarian value of the presented results. I have no doubt that this manuscript deserves to be published in its current form. I am not suggesting any further changes. The authors wrote their article very neatly and carefully.
Author Response
Thank you for the evaluation of our article. We are glad about your positive opinion.
Authors
Round 2
Reviewer 1 Report
Dear Authors,
Accept in present form.
Author Response
The authors are grateful for the comments.
Reviewer 2 Report
COMMENTS TO AUTHORS
Thank you for the revised version of your manuscript entitled: “Influence of harvesting time on the chemical composition of wild stinging nettle (Urtica dioica L.)“. You included important changes in the new text, and a new figure with data of light intensity during the experimental period, but you realize that you did not measured light intensity during your experiments and thus cannot provide any new data. In any case Fig. 2 gives important information of the environmental conditions in the experimental plot. The new paragraphs included in the text supply important information omitted in the first manuscript.
In your explanation letter you show information concerning the originality of a part of your data, which may be of interest for readers. In summary you significantly improved the manuscript. Please revise the English style in the text.
Author Response

(The authors gave the same response as above.)

Reviewer 3 Report
The authors have enhanced the manuscript, but English grammar needs to be improved before acceptance for publication. Nonetheless, the concentration of DPPH stock solution has not been mentioned in the experimental procedure (line 360). Moreover, the authors can describe the decades as ranges of years, in what concerns to the sunshine duration (lines 291-205).
Author Response
The authors are grateful to the reviewer for the observations.
The comments of the reviewer were taken into account and corrected.
- Nonetheless, the concentration of DPPH stock solution has not been mentioned in the experimental procedure (line 360).
In line 368 information were added: “….of 3 mL 0.1 mM of DPPH methanol solution…”
- Moreover, the authors can describe the decades as ranges of years, in what concerns to the sunshine duration (lines 291-205).
Lines 297-306. Taking into account the remark of another reviewer, the information on the sunshine duration (Fig. 2) was added.
To determine the effect of the sunshine duration on the number of pigments correlation analysis was performed.
Lines 183-184. However, the effect of decades and the total monthly sunshine duration on the number of pigments in the nettle leaves was not determined.
Lines 125-126, 202-203 The supplement information added on the influence of meteorological conditions on other parameters of nettle leaves AAC, TPC.
Taking into account comments on requiring English language editing, the language of the article has been edited.
Lines 91-94;
Line 98;
Lines 104-108;
Line 150;
Lines 152-153;
Lines 156-159;
Line 172;
Lines 190-191;
Lines 195-197;
Lines 201-202;
Lines 210-212;
Lines 215-252.